# Chitosan Nanoparticles Loaded with Polyphenols for Cosmeceutical Applications: A State-of-the-Art Review

**DOI:** 10.3390/pharmaceutics17081068

**Published:** 2025-08-18

**Authors:** Valeria Gaetano, Agnese Gagliardi, Elena Giuliano, Emanuela Longo, Donato Cosco

**Affiliations:** 1Department of Health Sciences, University “Magna Græcia” of Catanzaro, Campus Universitario “S Venuta”, 8100 Catanzaro, Italy; valeria.gaetano@unicz.it (V.G.); gagliardi@unicz.it (A.G.); elena.giuliano@unicz.it (E.G.); emanuela.longo@unicz.it (E.L.); 2“AGreenFood” Research Center, University “Magna Græcia” of Catanzaro, Campus Universitario “S Venuta”, 88100 Catanzaro, Italy

**Keywords:** polyphenols, cosmeceutical, chitosan, topical administration, skin diseases

## Abstract

Nanotechnology has been widely employed in the field of cosmeceuticals, promoting the development of innovative cosmetic formulations characterized by notable pharmacological activity. The use of nanocosmeceuticals allows for better skin penetration of active compounds, their controlled release over time, and greater physico-chemical stability. Chitosan nanoparticles have generated significant interest in the scientific community as dermal and transdermal delivery systems for natural compounds. In particular, the encapsulation of polyphenols within chitosan nanosystems has been proposed as a method to enhance the effectiveness of bioactives in cosmeceutical formulations. This review discusses the most relevant scientific literature on the topic, with particular attention to studies published in recent years. Chitosan-based nanosystems improve the stability, bioavailability, and skin compatibility of polyphenols, offering promising solutions for the prevention and treatment of skin disorders due to their antioxidant and anti-inflammatory properties. This review provides a comprehensive update on the development of chitosan nanoparticles containing polyphenols and their potential clinical applications, highlighting the role of these systems as nanocosmeceuticals.

## 1. Introduction

There is a bridge between cosmetics and pharmaceuticals, which allows for the discussion of cosmeceuticals products valued by many consumers for both their aesthetic and therapeutic properties that can be used to treat various disorders of the skin, hair, nails, and oral cavity [1,2].

The term “cosmeceutical” was first coined by R.E. Reed in 1962. Today, it carries a different meaning, established by Dr. Albert Kligman of the University of Pennsylvania in 1984. According to Dr. Kligman, a cosmeceutical is “a topical preparation that is sold as a cosmetic but has performance characteristics that suggest pharmaceutical action” [3]. The cosmeceutical sector has grown exponentially, as these formulations may offer effective solutions to common skin disorders such as photoaging and hyperpigmentation, providing anti-aging and moisturizing effects. Additionally, they can be used for the treatment of infections and inflammatory diseases, thanks to components with antioxidant and/or anti-inflammatory properties (Figure 1) [4,5]. Therefore, they represent a new generation of cosmetic products that contain active ingredients capable of improving not only the appearance but also the health of the skin.

The visual appearance of acne, the scars it leaves, and the presence of spots or wrinkles are considered significant inconveniences for many individuals, with a considerable social impact that affects quality of life [1]. Tools like the Baumann Skin Type Indicator (BSTI) questionnaire are used to identify patients at higher risk of premature aging or hyperpigmentation [6]. Aging is a multifactorial process, genetically driven and influenced by environmental and behavioral factors such as smoking, pollution, stress, and poor nutrition. In particular, chronic exposure to UV radiation has been shown to cause DNA damage and the formation of free radicals, leading to oxidative stress and tissue inflammation [7]. This results in the degradation of collagen and elastin fibers, key components for maintaining skin elasticity and integrity, leading to the formation of wrinkles and solar elastosis, which is characterized by the abnormal accumulation of elastin in the skin [8,9]. These issues can be addressed by using formulations that stimulate collagen production and counteract free radical damage while maintaining the structure of keratin [10]. Aesthetic medicine is often used to correct visible signs of aging through cosmetic surgery and noninvasive therapies, such as fillers and lasers. However, skincare products remain the most accessible option for the majority of people [11].

According to Russell-Goldman and Murphy, skin aging goes beyond being a mere aesthetic concern and should be considered a genuine disease [12]. As skin reflects deeper tissue aging, several studies comparing sun-protected and aged skin have shown dysregulation in gene expression related to cytoskeletal integrity, immune function, and metabolism [13,14,15]. Moreover, the skin acts as a “neuro-endocrine-immune organ,” playing a key role in systemic homeostasis. In pathological conditions such as atopic dermatitis, excessive production of inflammatory cytokines can trigger a state of chronic inflammation, increasing the risk of type 2 diabetes and cardiovascular disease [12]. In this context, the topical administration of polyphenols encapsulated in chitosan nanoparticles emerges as an innovative and targeted therapeutic strategy, capable of locally modulating the cutaneous inflammatory response and preventing its systemic implications. Therefore, preventing skin senescence and the associated age-related dysfunctions can benefit the health of internal organs [16]. Conventional topical formulations are currently the first-line treatment for managing mild skin conditions [17,18,19]. However, although conventional topical therapy can reduce symptoms, it has several limitations, such as the difficulty of active compounds effectively penetrating the epidermal barrier, which remains the primary obstacle to absorption. Other limitations include the poor concentration of the drug(s) at the target site, potential adverse effects, and the need for prolonged administration regimens before visible effects are achieved [20].

Considering these factors, research has focused on the development of innovative nanotechnology-based drug delivery systems designed to enhance the skin permeation of active compounds [5]. These nanoformulations offer several advantages over conventional systems, including greater stability of the entrapped compounds, the ability to exploit site-specific targeting, high encapsulation efficiency, controlled and long-lasting drug release, lower side effects, and enhanced skin permeation profiles [21]. The physicochemical properties of the nanomaterials influence the interactions between the nanoparticles and the skin, enabling the nanosystems to reach specific layers of the skin in sufficient amounts to produce the desired effects [5]. Several drug delivery systems have been proposed as cosmeceuticals, including liposomes, ethosomes, transfersomes, solid lipid and polymeric nanoparticles, nanoemulsions, gold nanoparticles, fullerenes, dendrimers, and cubosomes [22]. The scientific literature has shown that polymer-based nanoparticles can help reduce the degradation of active compounds, minimize their potential side effects, promote controlled release, and enhance skin penetration [23]. Chitosan-based nanoparticles have been proposed as carriers of several therapeutic agents, including those intended for topical administration due to their unique properties, such as biocompatibility, biodegradability, and the ability to retain various bioactives [24,25,26,27]. Thanks to its cationic nature, chitosan promotes skin adhesion and penetration through the epidermal barrier, thereby enhancing the bioavailability of active compounds [28]. Chitosan can undergo chemical modifications, such as carboxymethylation and quaternization, to optimize its molecular weight, degree of deacetylation, and hydrophilic–lipophilic balance, thus improving its performance as a main component of a delivery system [29]. Recent studies have shown that such modifications enhance the solubility, stability, and therapeutic efficacy of the encapsulated compounds [30,31,32]. Nasr et al. also highlighted the potential of chemically modified chitosan derivatives in enhancing the transdermal delivery of bioactive compounds. In particular, they developed nanoemulsions coated with amphiphilic oligochitosan for the topical delivery of *Thymus vulgaris* essential oil. Their findings demonstrated improved physical stability, enhanced skin permeation, and superior anti-inflammatory and anticancer effects, confirming the versatility of chitosan-based carriers for the dermal delivery of natural agents that are often unstable and poorly permeable across the skin barrier [33]. Furthermore, the encapsulation of polyphenols in chitosan nanoparticles helps to preserve their biological activity, protect them from oxidation, and enable their sustained release, thereby reducing the administration times and the risk of skin irritation [34]. These nanoparticles are also characterized by a high stability, preventing phase separation and ingredient degradation [35]. Chitosan is considered a multifunctional smart material, also known for its antimicrobial, antioxidant, antibacterial, and hemostatic properties, as well as for its ability to form bonds with crosslinking agents, which promote the entrapment of active compounds in the polymeric matrix and modulate their release over time [36,37]. In recent years, the topical administration of polyphenols through innovative nanoformulations has garnered increasing scientific interest due to the remarkable pharmacological properties of these molecules [38].

While previous reviews described the individual or combined use of chitosan nanoparticles and polyphenols in pharmaceutical and biomedical contexts, a focused analysis of their association to develop cosmeceutical formulations is still lacking [39,40]. This study fills that gap by providing an updated and detailed overview of various technological strategies, molecular interactions, and experimental evidence regarding the effectiveness of chitosan-based nanosystems in enhancing the therapeutic outcomes and cosmetic properties of polyphenols. This review seeks to provide the state of the art of the latest developments in this field, with a particular focus on studies published between 2018 and 2024, highlighting recent advances and emerging trends in the use of chitosan nanosystems for the delivery of polyphenols in cosmeceutical applications. Furthermore, it discusses future perspectives about the potential application of these formulations in regenerative medicine and advanced dermatology (Figure 2).

## 2. Skin as a Biobarrier

The absorption of active compounds and nanosystems through the skin can occur via transcellular, intercellular, or transfollicular routes. The first two involve the passage of molecules through the stratum corneum, while the third relies on the movement of molecules through sweat glands and hair follicles (Figure 3). Hair follicles have been shown to play a critical role in enhancing the dermal delivery of active compounds, especially when nanosystems are employed [41]. These complex structures extend deep into the skin, up to several millimeters, and comprise over 20 different cell populations, including melanocytes, endothelial cells, immune cells, and stem cells [42,43].

Several in vitro and in vivo studies have shown that follicles act as true reservoirs for nanocarriers, enabling a prolonged release of encapsulated drugs. This mechanism contributes to improved local bioavailability and may help reduce systemic side effects [44,45]. As illustrated in Figure 3, the transfollicular route bypasses the densely packed corneocytes of the stratum corneum, providing a more direct and efficient pathway for certain formulations, especially in the case of nanoparticle-based delivery systems [46]. Moreover, the dense capillary network surrounding the follicle supplies blood flow that facilitates systemic absorption of drugs accumulated in the deeper follicular regions. This makes hair follicles not only important local reservoirs but also potential gateways for transdermal drug delivery [47].

A practical example of this behavior is shown in a study comparing the penetration of a fluorescent dye, encapsulated in poly(D,L-lactide-co-glycolide) nanoparticles versus its free form, into the hair follicles of porcine skin. The study showed that nanoencapsulation promotes enhanced passage, especially after the application of a massage [48]. The cuticles along the hair shaft help move particles deeper into the hair follicle, which acts as a reservoir for up to 10 days. In contrast, free molecules cannot be stored in this compartment, as they are primarily localized in the outer skin layers, which undergo desquamation [48]. On the other hand, several nanocarriers can cross the stratum corneum and release the encapsulated active compounds [49].

The intercellular and intracellular pathways of skin penetration exhibit substantial differences that critically influence the design of delivery systems [50]. The intercellular route is the most common and takes place in the extracellular space between corneocytes, where the lipid matrix represents the main barrier to diffusion. For this reason, materials intended to exploit this pathway must be characterized by high lipophilic properties or possess features that allow favorable interaction with epidermal lipids, such as lipophilic coatings or structural modifications that improve compatibility with the lipid phase. On the contrary, the intracellular route requires the direct trespass of cell membranes, involving passage through alternating hydrophilic and lipophilic environments [51,52]. In this case, the material design focuses on surface properties such as charge or the presence of specific functional groups. Nanocarriers with a certain degree of amphiphilicity may be good candidates to overcome this barrier [52]. Understanding these differences enables the rational selection of key parameters, such as mean sizes, charge, lipophilicity, and functionalization, during development, optimizing the effectiveness of the delivery system according to the selected penetration pathway [53].

Skin penetration tests are crucial for evaluating the performance of topical or transdermal formulations. Nanoparticles must be able to cross the skin barrier, release the entrapped compound contents, and degrade without causing side effects. Human skin obtained from autopsies or plastic surgery is considered the most suitable model for ex vivo skin penetration studies. However, animal skin (pig, mouse, rat, and rabbit) is often used for experimental investigations, despite structural and morphological differences between species that can influence the results. These differences include skin thickness, the number of hair follicles per area, and the total amount or composition of the lipid matrix surrounding the corneocytes [54]. Tests can be performed using epidermal membranes, dermatomal skin, or full-thickness skin. While the epidermis provides more reliable information, it requires proper preparation, heating, or chemical treatment, which can cause structural damage [55]. Pig skin is used to evaluate drug penetration through hair follicles due to its similarity to human skin [53]. In vitro results are influenced by the method or apparatus used (e.g., Franz diffusion cells) and the skin condition. According to OECD guidelines, skin penetration tests should be performed using dermatomized skin with a thickness of 200 to 400 μm [53]. In addition, the use of reconstructed human skin models and synthetic membranes offers a better alternative for achieving reproducibility of results. Other important factors to consider include tissue preservation and the influence of the physico-chemical properties of nanoparticles and their interaction with the skin [53].

The size of nanoparticles plays a crucial role in their ability to deliver bioactive compounds through the skin barrier [52]. Previous studies have shown that this parameter differently influences their skin penetration: small particles (~80 nm) can cross the viable epidermal layer and accumulate in hair follicles, while larger particles (~500 nm) mainly move along the hair follicles without deep penetration. Intermediate sizes (~200 nm) exhibit moderate effectiveness in delivering active compounds. These findings suggest that a precise control of the size of chitosan nanoparticles is essential to optimize the bioavailability and therapeutic efficacy of topically applied polyphenols [33,48,56].

## 3. The Role of Polyphenols in Skin Health

Currently, there is growing interest in cosmetics made from natural compounds, driven by the desire to safeguard health and avoid the potential side effects associated with synthetic products, such as skin irritations, phototoxicity, and mutagenicity. Additionally, using renewable and biodegradable natural resources helps preserve nature [57]. The global natural cosmetics market is estimated to reach USD 37.9 billion in 2023, with a compound annual growth rate (CAGR) of over 5.1% from 2024 to 2032 (https://www.gminsights.com). Recent experimental studies have shown that the increasing use of plant by-products rich in polyphenols as natural ingredients offers beneficial effects for human health [58].

The use of polyphenols or plant extracts enriched with these bioactives has been widely proposed for cosmeceutical applications due to their unique biological properties, such as antioxidant, anti-inflammatory, antimicrobial, anti-aging, and sun protection characteristics [59]. Moreover, these compounds show significant effects against the progression of various skin diseases, including wounds and burns [60]. Additionally, they help prevent photocarcinogenesis and promote hair growth [61]. Polyphenols are a class of secondary metabolites of plants characterized by defensive properties against external aggressors and the ability to confer color [62]. Their synthesis is linked to the pentose phosphate, shikimate, and phenylpropanoid pathways [63,64]. With the development of science and technology, more than 1000 types of polyphenols have been identified and their pharmacological activities investigated, which are associated with their chemical structure. They are usually classified into four classes/families as a function of the number of phenolic rings they contain and their different chemical structures: flavonoids, stilbenes, lignans, and phenolic acids [62] (Table 1). Flavonoids, which share a common structure consisting of two aromatic rings linked by three carbon atoms that form an oxygenated heterocycle, can be divided into different subclasses based on the type of heterocycle involved: flavonols, flavones, isoflavones, flavanones, anthocyanins, and flavanols [62]. Phenolic acids can be categorized into benzoic acid derivatives and cinnamic acid derivatives [65]. These compounds interact with the skin barrier as a function of their physico-chemical properties. In particular, the skin permeation of polyphenols depends on several factors, including molecular weight, lipophilicity, and their ability to interact with skin proteins [66]. Studies have shown that low molecular weight polyphenols, such as protocatechuic acid and catechin, tend to have greater solubility in lipid environments, demonstrating a higher ability to penetrate the skin barrier compared to more complex compounds like quercetin and rutin. Moreover, their interaction with the protein of the stratum corneum can influence the lipid structure of the skin, increasing permeability and enhancing the absorption of other bioactive compounds [67]. A wide variety of plant polyphenols are used in cosmeceutical products for their antioxidant and anti-inflammatory properties. The presence of aromatic rings, highly conjugated residues, and hydroxyl groups enables these compounds to act as effective electron or hydrogen atom donors, neutralizing free radicals and preventing oxidative stress induced by UV radiation [68]. This property helps prevent the signs of premature aging. Additionally, various polyphenols can effectively reduce inflammatory skin conditions such as acne or dermatitis [69]. Furthermore, they have shown other beneficial properties, including the ability to inhibit gene expression and the activity of skin enzymes, such as hyaluronidase, matrix metalloproteinase (MMP), collagenase, and serine protease elastase [70], as well as improve the wound healing process [71].

### 3.1. Antioxidant Activity

When human skin is overexposed to UV light, it triggers the production of reactive oxygen species (ROS), which cause oxidative stress [72]. This stress contributes to the onset of skin disorders such as hyperpigmentation, premature aging, and dry skin. Furthermore, the pathophysiology of skin cancers, inflammatory diseases, and allergic conditions is also influenced by ROS [73].

The skin, particularly the dermis, contains a sufficient number of antioxidants and defense mechanisms against free radicals and radical reactions. Studies conducted on living organisms have shown that, following photo- or chrono-aging, the skin undergoes alterations in both enzymatic and antioxidant components [74]. As previously discussed, polyphenols prevent lipid peroxidation and can decrease free radicals. Substances rich in polyphenols, such as grapes and green tea, are commonly used in topical formulations to prevent skin aging [75]. Several experimental studies have shown antioxidant and anti-aging effects of resveratrol on the skin [76,77,78]. Its mechanism of action involves activation of the Nrf2 pathway and inhibition of NF-κB and AP-1, as well as the upregulation of heat shock protein 27 (HSP27), contributing to reducing the oxidative stress and inflammation [79]. Studies have demonstrated that resveratrol stimulates fibroblast proliferation and enhances the production of type I, II, and III collagen via activation of the anti-aging factor sirtuin 1 (SIRT1), thereby improving skin firmness, elasticity, and reducing pigmentation [80]. One study showed a significant improvement in wrinkles, skin elasticity, hyperpigmentation, and dermal thickness after daily applications of a topical mixture containing resveratrol, baicalin, and vitamin E for 12 weeks, while another study reported a 20% increase in hydration after just 2 weeks of treatment with resveratrol [81,82,83]. These findings support its efficacy as a multifunctional compound in cosmeceutical applications. Similarly, grape by-products, typically rich in anthocyanins, have been tested in vitro on human keratinocytes (HaCaT cells) for their antioxidant activity and have been proposed as potential materials for use in the cosmetic industry [84].

The potential anti-aging effect of an artichoke extract, rich in various polyphenols, including hydroxycinnamic acids and flavonoids, has been investigated by evaluating its antioxidant and anti-inflammatory effects [85]. Specifically, it was shown that the extract can improve skin elasticity and reduce roughness by inhibiting vascular aging, enhancing the function of endothelial and lymphatic vessels as a result of their protection from ROS-induced oxidative damage [86]. It has also been shown to increase the expression of genes involved in anti-aging mechanisms and improve cell cohesion by strengthening the formation of tight junctions, which are crucial for maintaining the structure of the skin barrier. Additionally, chlorogenic acid in the extract has shown free radical scavenging and UV protective activity [87]. The extract was tested in vitro and in vivo on 20 volunteers aged between 35 and 55 with sagging facial skin. They were treated with a cream containing 0.002% extract on one half of the face for 28 days, while a placebo cream was applied to the other half. The results showed that the artichoke extract improved skin elasticity and reduced roughness compared to the placebo [88].

The presence of polyphenols in extracts from various citrus species, especially red orange, has been shown to protect and prevent UV-induced damage in fibroblasts and keratinocytes. Citrus lemon peel extracts have also been tested on keratinocyte cells, demonstrating significant protection against oxidative stress due to the activity of gallic acid, catechin, and caffeic acid [89].

The polyphenols in green tea have exhibited protective effects against UV-induced skin damage. Most of these polyphenols are monomeric flavonoids known as catechins, which make up 75% of the total amount of these compounds [90]. Catechins provide numerous health benefits by scavenging free radicals and delaying the degradation of the extracellular matrix induced by ultraviolet radiation [91]. The main catechins are (-)-catechin (C), (-)-epicatechin, (-)-epigallocatechin (EGC), (-)-epicatechin-3-gallate, (-)-epigallocatechin-3-gallate (EGCG), and (-)-gallocatechin-3-gallate [92]. EGCG is the primary catechin and has been the most extensively studied in terms of its therapeutic properties on human skin, known for its ability to counteract UVB-induced skin damage by modulating key signaling pathways in keratinocytes and fibroblasts [93]. These mechanisms include inhibition of the epidermal growth factor receptor and suppression of pro-inflammatory mediators such as NF-κB, AP-1, TNF-α, IL-1α, IL-6, and MMP-1, all of which are involved in skin photoaging [93,94]. An in vivo study carried out on 20 volunteers aimed to investigate whether formulations containing 2% and 3% green tea extract (GTE) could be effective against skin photoaging and photoimmunological phenomena [95]. In detail, volunteers were exposed to UV radiation on the upper back at a minimum dosage of 1.5 minimal erythema doses (MED) per day for 4 days. Sunscreens containing different concentrations of GTE provided significant protection against photoaging and photoimmunology-related phenomena [95]. In another experimental study, green tea extracts were shown to reduce the number of sunburned cells and protect Langerhans cells in the epidermis from UV-induced damage [96].

In addition, tannase treatment, which cleaves the ester bonds of EGCG and ECG present in green tea, has been shown to enhance the antioxidant activity of green tea extract due to an increased concentration of GA, EGC, and EGCG, thereby providing excellent anti-wrinkle effects [97].

### 3.2. Anti-Inflammatory Activity

Inflammatory dermatoses include several types of skin disorders, such as atopic dermatitis, psoriasis, and urticaria [98,99]. The damaged stratum corneum releases signaling molecules that trigger cytokine cascades, leading to inflammation. Conventional therapies include glucocorticoids and biologic agents, but both can only be used for short periods due to the risk of serious side effects [100].

In this context, polyphenols have emerged as promising natural anti-inflammatory agents. Compounds such as resveratrol, chlorogenic acid, caffeic acid, curcumin, and rutin exert their effects by modulating the expression of inflammatory genes, notably through the inhibition of COX-2 and iNOS enzymes. Experimental studies have shown that orange peel extracts can downregulate COX-2 expression and reduce PGE2 production via activation of PPAR-γ in keratinocytes exposed to UVB [101]. Similarly, rutin application in UVB-irradiated mice reduced epidermal hyperplasia and oxidative damage [102]. Oleuropein and hydroxytyrosol, have demonstrated positive effects against senescence processes in human lung and dermal fibroblasts [103]. Galanakis et al. investigated the application of different concentrations of hydroxytyrosol, tyrosol, and oleuropein, recovered from mill wastewater, as UV filter boosters, highlighting their potential use in sunscreen formulations [104]. Moreover, a bergamot polyphenolic fraction (BPF), tested on keratinocytes after UVB exposure, promoted recovery of cell viability through modulation of the pro-inflammatory cytokine IL-1β [105]. Additionally, BPF treatment has been shown to restore telomere length and activity [106]. Interestingly, carnosol, an active compound found in rosemary, reduced inflammatory markers in mice and has proven effective in the treatment of atopic dermatitis lesions [107].

### 3.3. Antimicrobial Activity

The rise in antibiotic resistance has driven research efforts to explore potential antimicrobial derivatives of natural origin [108]. Flavonoids act on the membrane or cell wall of microorganisms, causing structural and functional damage [109]. Hydrophobic flavonoids can insert themselves into the hydrophobic layer of the membrane, while hydrophilic ones interact with the polar groups of lipids through hydrogen bonding [110]. Significant antibacterial activity of catechins has been shown against both Gram-positive and Gram-negative bacteria, owing to their ability to interact with the membrane, compromising its integrity or inducing fusion [110]. Flavonoids can also inhibit nucleic acid synthesis by intercalating between DNA bases and inhibiting DNA gyrase [111]. Among these, quercetin, apigenin, and sakuranetin have been demonstrated to inhibit the bacterial enzyme β-hydroxyacyl-acyl dehydratase, essential for lipid biosynthesis in *Helicobacter pylori*, thereby compromising the bacterial growth and homeostasis [112]. Quercetin also induces DNA cleavage, and it interferes with membrane potential, further impairing bacterial survival [113]. Another important property of flavonoids is their ability to inhibit biofilm formation, an essential defense mechanism of bacteria. They do this by promoting cell aggregation, interfering with the nutrient uptake and preventing biofilm maturation [114].

Different types of infections can affect the skin, leading to various dermocosmetic issues such as acne. Acne can be caused by multiple mechanisms, including increased sebum production by the sebaceous glands, altered keratinization, inflammation around the pilosebaceous follicles, and the proliferation of *Propionibacterium acnes* [115]. Tea polyphenols have shown promise as molecules for the treatment of acne vulgaris due to their ability to reduce sebum production in the skin while also acting as anti-inflammatory and antimicrobial agents [116]. In an experimental study performed in Pakistan, 22 healthy, non-smoking men aged 22 to 28 were enrolled to evaluate the effectiveness of a topical formulation of lotus and green tea on facial sebum production [117]. The volunteers were divided into two groups: the first group applied topical green tea to one cheek and a placebo to the other. The second group applied a combination of lotus and green tea to one cheek and a placebo to the other. Sebum secretion was measured using a sebometer, and the results showed a significant reduction in sebum production in both groups after 60 days. Specifically, there was a 25% decrease for the first group and a 27% decrease for the second, confirming the efficacy of polyphenol-based cosmetics for treating skin disorders [118].

## 4. Limits of Polyphenols and Innovative Strategies of Administration

Polyphenols have been widely studied as ingredients in cosmeceutical formulations due to their previously described unique properties. However, their biological activity in the skin is primarily influenced by their physicochemical characteristics and their ability to cross the epidermal barrier to reach the target compartment [119]. Several factors limit their use, including variable solubility, reduced skin permeability, and instability to heat, pH, and oxygen, which promote their degradation [39]. Despite their sun protection capabilities, some of these compounds have been shown to be photounstable after prolonged exposure to UV rays, leading to phototoxic reactions [120].

In recent years, nanotechnology has been employed in cosmetology and dermatology, leading to the development of several nanocarriers aimed at enhancing the effectiveness of active compounds [121]. This approach has helped overcome the limitations of polyphenols by modulating their solubility, physicochemical stability, bioavailability, and skin permeation and promoting their prolonged release over time [122].

The main challenge of a drug delivery system designed for skin application is to bypass the stratum corneum. This is the largest biobarrier of the body [123]. Polymeric nanoparticles can be classified based on morphology, average size, composition, and physico-chemical properties [124]. Their surface can be modified to enhance interaction with the skin, facilitating absorption [125]. As a result, they represent a promising strategy to improve the efficacy and safety of encapsulated compounds. This paves the way for new cosmetic and therapeutic formulations for skin application and treatment of dermatological diseases [126].

## 5. Chitosan in Cosmetics and Cosmeceuticals

Currently, natural polymers primarily used in the preparation of polyphenol-loaded nanodelivery systems include proteins and polysaccharides [127]. Among the most abundant polymers in nature, polysaccharides are inexpensive and readily available, making them valuable biomaterials for the development of innovative formulations. Chitosan is a versatile polysaccharide, often referred to as the biopolymer of the twenty-first century due to its unique properties. Its use has grown exponentially over the last decade [99]. Specifically, its biocompatibility, non-toxicity, and biological activities, including antioxidant, anti-inflammatory, antimicrobial, and wound healing effects, have made it highly attractive in biomedical, pharmaceutical, and cosmetic fields [128,129].

In cosmetics, it acts as both an active ingredient and carrier material, delivering key benefits such as moisturization, which forms a hydrophilic film on the skin to prevent water loss and protect and improve skin and hair affinity due to its interaction with keratin [130,131]. It is especially promising for the development of formulations proposed for the treatment of acne, psoriasis, hyperpigmentation, and post-surgical scarring, and it is also widely employed in film-forming applications such as wound dressings and protective skin layers [132,133].

## 6. Chitosan Nanoparticles as Carriers of Polyphenols for Topical Applications

### 6.1. Physico-Chemical Properties of Chitosan

Chitosan is a cationic, highly basic, and mucoadhesive polysaccharide widely used for the micro- and nanoencapsulation of bioactive compounds [134,135,136]. It is derived from chitin, a natural polymer found in the exoskeletons of crustaceans, mollusks, and insects and in the cell walls of fungi [4]. Chitin consists of N-acetyl-D-glucosamine and D-glucosamine units linked by β-1,4-glycosidic bonds [137].

Chitosan is produced through a two-step process: an initial purification step, followed by a deacetylation reaction. The purification of chitin involves an acid treatment with 1 M hydrochloric acid (HCl) to remove minerals, followed by a basic treatment with 1 M sodium hydroxide (NaOH) to eliminate proteins. Once purified, chitin is subjected to a deacetylation reaction, which involves treating the material with concentrated NaOH (40–50%) at high temperatures (80–100 °C) for 2–6 h. This step enables the removal of at least 60% of the acetyl groups from the N-acetyl-D-glucosamine units, converting them into D-glucosamine units and exposing free amino groups responsible for the peculiar cationic and pH-sensitive properties of the polysaccharide (Figure 4) [138,139,140]. Under acidic conditions, especially at pH values below its pKa (~6.3), the amino groups become protonated, increasing the solubility in water of the polymer. Conversely, at higher pH values, deprotonation reduces its dissolution capacity. The operating conditions of the deacetylation step significantly influence the physico-chemical properties of the chitosan. In particular, harsher treatment conditions can increase the degree of deacetylation, thereby enhancing the positive charge of the polymer; however, they may also promote chain scission, leading to a decrease in molecular weight [141,142]. This aspect is crucial, as molecular weight directly affects the viscosity, film, gel-forming abilities, and mechanical stability of chitosan. High molecular weight derivatives are usually characterized by greater structural stability and superior film-forming properties, making them ideal for applications such as scaffold development in tissue engineering and controlled drug release. On the other hand, low molecular weight chitosan exhibits higher solubility and permeability, making it more suitable for specific applications such as transdermal or ophthalmic drug delivery [143,144].

Furthermore, the presence of free amino groups endows chitosan with strong chelating capabilities, enabling selective interaction with metal ions [145]. This property is particularly advantageous for wastewater treatment, especially in the removal of heavy metals, and is also highly relevant in the biomedical and pharmaceutical fields. In these areas, the metal-binding ability of chitosan can be employed to improve the drug delivery efficiency and to protect various bioactive compounds from degradation [145,146].

### 6.2. Enhancement of Bioavailability and Skin Permeability by Chitosan Nanoparticles

The use of polymeric nanoparticles, especially those based on chitosan, represents an innovative and highly effective strategy to enhance the bioavailability and cutaneous permeation of bioactive compounds [124,125]. These nanosystems are widely studied in pharmaceutical and cosmeceutical applications due to their ability to facilitate the passage of active ingredients through the skin barrier as well as their tendency to localize within hair follicles, where they act as reservoirs for the controlled and gradual release of active compounds [147].

The cationic nature of chitosan plays a crucial role in enhancing skin permeability, especially in the case of hydrophobic molecules. The protonated amino groups under physiological pH confer a positive charge that promotes strong electrostatic interactions with the negatively charged components of the stratum corneum, such as lipids and proteins [148,149]. As a result, chitosan nanoparticles are able to efficiently interact with the skin, enhancing adhesion and prolonging the residence time of active compounds on the skin surface [149].

In addition, chitosan can alter the morphology of the stratum corneum by disrupting the organization of intercellular lipids and temporarily weakening the tight junctions between corneocytes [150]. This effect increases skin hydration, reduces cellular cohesion, and leads to a reversible modification of the skin barrier function, thus facilitating the diffusion of active compounds through the epidermis [151]. Another relevant mechanism involves the modulation of tight junctions in the upper layers of the epidermis, which promotes paracellular permeation, particularly beneficial for macromolecules and hydrophilic compounds [152].

Its pH-sensitive nature causes the swelling or shrinking of nanoparticles as a function of the skin environment, helping to modulate the release of encapsulated drugs [153]. Chitosan-based formulations generally demonstrate prolonged release and improved local bioavailability, maintaining more stable therapeutic concentrations and reducing the administration times of bioactives [154]. Finally, the encapsulation of active compounds within nanoparticles protects them from environmental and metabolic degradation, helping preserve their therapeutic efficacy over time [33].

Polyphenols encapsulated in chitosan nanoparticles exhibit enhanced biological activity due to prolonged release and an improved interaction with skin layers [28]. For example, Santana Gomes et al. demonstrated in both animal and human models that chitosan NPs increase skin permeability, enable controlled release, and improve tolerability compared to free compounds. These formulations have been demonstrated to be efficacious in the treatment of psoriasis, dermatitis, wounds, and melanoma, yielding therapeutic benefits with minimal side effects [155].

### 6.3. Preparation Methods of Chitosan Nanoparticles

Several approaches have been proposed for the preparation of chitosan nanoparticles, including microemulsion, solvent diffusion emulsification, polyelectrolyte complexation, complex coacervation, and solvent evaporation [156]. Each of these methods offers specific advantages but also limitations that affect their applicability in cosmetic and pharmaceutical applications. Among them, ionic gelation is the most commonly employed technique. It involves crosslinking chitosan with polyanionic agents such as sodium tripolyphosphate (TPP), leading to the formation of stable chitosan–TPP nanoparticles [157]. Polyphenols can be entrapped into the chitosan matrix before the addition of TPP, allowing their encapsulation through various mechanisms, including hydrogen bonding, electrostatic interactions, and hydrophobic forces. This process generates nanosystems with high biocompatibility, improved colloidal stability, and efficient retention of polyphenolic compounds (Figure 5) [158]. This method is advantageous due to its simplicity, speed, and the ability to be carried out at room temperature, thereby avoiding thermal stress that could compromise the chemical structure and biological activity of these compounds. Moreover, the absence of toxic organic solvents or harsh chemical minimizes the risk of molecular degradation, ensuring a high level of safety and biocompatibility, essential characteristics for formulations proposed for topical applications on skin and mucous membranes [159]. Another advantage is related to the precise control over particle size, generating systems with low polydispersity and good colloidal stability. Furthermore, the positive surface charge conferred by the protonated amino groups of chitosan generates electrostatic repulsion between nanoparticles, which helps maintain a stable dispersion over time by preventing aggregation and sedimentation [159]. This is particularly relevant because the average size of nanoparticles directly influences the skin permeability: while large particles tend to remain on the skin surface, nanosystems with an average diameter equal to or less than 300 nm are able to penetrate the deeper layers of the epidermis, significantly enhancing the efficacy of delivered compounds [160].

Other methods present some drawbacks: polyelectrolyte complexation, although free of organic solvents, can be less reliable in controlling nanoparticle size and stability [161]. Emulsification and solvent evaporation techniques require organic solvents and surfactants, which may be potentially irritating or toxic to the skin, and offer less precise size control. The microemulsion approach produces very small nanoparticles (below 100 nm) but often needs organic solvents and longer processing times [156,162]. Chemical crosslinking uses agents like glutaraldehyde that may leave undesirable residues, compromising biological properties and making it less suitable for cosmetic use [163]. Finally, complex coacervation carried out in an aqueous environment at low temperatures is high-pH sensitive, inducing potential destabilization or ineffective encapsulation of bioactives [164] (Table 2).

### 6.4. Applications of Chitosan Nanoparticles for Specific Polyphenol Classes

#### 6.4.1. Flavonoids

Ex vivo permeation studies have demonstrated that chitosan nanoparticles, with an average diameter ranging from 30 to 200 nm, act as effective carriers for the transdermal delivery of hydrophobic drugs such as curcumin. Their ability to be accumulated in hair follicles facilitates the delivery of the active compounds into the deeper layers of the skin, as further confirmed by confocal laser scanning microscopy analyses [165,166]. These nanosystems significantly increased the transdermal drug flux, achieving values of 5.14 ± 1.31 μg/cm^2^/h compared to conventional formulations [165].

Curcumin is a natural compound known for its various pharmacological properties, particularly in the cosmeceutical field. It has shown efficacy in treating conditions such as psoriasis, hypertrophic scars caused by collagen overproduction, and vitiligo [167]. Additionally, curcumin promotes wound healing due to its ability to enhance the spread of collagen–chitosan scaffolds (Table 3) [165]. Curcumin-loaded chitosan nanoparticles represent an innovative system for enhancing the skin permeability and transdermal bioavailability of the active compound [168]. Nair et al. demonstrated that curcumin-loaded chitosan nanoparticles exhibit a high entrapment efficiency (more than 80% when 0.3 mg and 1.5 mg of active compound have been used during the sample preparation), a prolonged leakage of the polyphenol, and an enhanced skin permeation with respect to aqueous solutions of the bioactive [169]. Furthermore, the integration of these nanoparticles into a keratin–chitosan gel enhanced their skin retention, enabling a prolonged release and a greater therapeutic efficacy of the formulations [170].

Quercetin, a flavonoid renowned for its antioxidant and anti-inflammatory properties, has been successfully encapsulated in lecithin–chitosan nanoparticles (NPs) containing D-α-tocopheryl polyethylene glycol 1000 succinate (TPGS) as a surfactant, with the aim of enhancing its skin absorption [171]. These nanoparticles, obtained by electrostatic interactions between positively charged chitosan and negatively charged lecithin residues, were characterized by a phospholipid-based hydrophobic core surrounded by a hydrophilic shell of the polysaccharide. The hydrophobic residues of TPGS and lecithin interact to form the core where lipophilic quercetin is dissolved and protected by the chitosan shell [171,172]. The addition of TPGS to the nanosystem resulted in an increase in the entrapment efficiency of quercetin, particularly at a surfactant concentration of 2% *w*/*v*. The nanoparticles were prepared using gelling techniques and were characterized by a mean diameter of 100–150 nm and a relatively high polydispersity index (0.3–0.4), likely due to the micelles formed by TPGS [171]. The zeta potential of the NPs containing quercetin was lower than that of the empty systems, likely due to the electrostatic attraction between the –OH residues of the active compound and the positively charged biopolymer. In vivo studies showed that the NPs significantly promoted the accumulation of quercetin in the epidermis, confirming the suitability of these nanosystems for the skin delivery of quercetin and highlighting their potential to preserve its antioxidant and anti-inflammatory effects. In addition, the hydrated external layer of chitosan enhances skin wetting, which could help reduce or prevent dehydration. Mice treated with quercetin-loaded NPs exhibited a thicker stratum corneum, a reduced number of cell junctions, and an increase in intercellular space, leading to improved skin permeation of the bioactive compound. Furthermore, in vitro tests showed that the nanoparticles provided significant protection against UVB radiation-induced damage [173]. In another experimental study, quercetin-loaded chitosan nanoparticles provided effective skin protection against UVB radiation-induced damage [174]. The nanosystems, characterized by a mean diameter of 184 nm and a positive zeta potential, enhanced the efficacy of quercetin in inhibiting the NF-κB/COX-2 signaling pathway, thereby reducing skin edema [172]. A subcategory of flavonoids, known as flavanols, includes catechins and epicatechins, which are found in green tea, cocoa, and certain fruits [175]. Green tea extracts, rich in polyphenols, are used in cosmetic formulations; however, catechins demonstrated limited skin permeability, attributed to their hydrophilic nature and chemical interactions with lipid bilayers. Among them, only EC and EGC are able to permeate the skin, while EGCG and ECG show only a limited penetration [176]. However, nanotechnology can modulate this trend: liposomes were used to entrap EGCG, but its release from the vesicles was limited due to the binding between the active compound and the lipid bilayers, whereas chitosan nanoparticles proved to be an ideal carrier [177,178]. In detail, the nanosystems were used for the treatment of psoriasis induced by imiquimod in mouse models [179]. EGCG has shown good anti-inflammatory effects; however, its bioavailability is limited, which compromises its use for the treatment of psoriasis. The encapsulation within chitosan nanoparticles, however, improved skin lesions and epidermal architecture while reducing TPA- and IL-22-induced inflammation in keratinocytes at a concentration four times lower than its free form. Moreover, histological analysis showed a significant reduction in cell hyperproliferation [180].

Although limited research has been conducted on the treatment of cellulite using nanosystems, an effective formulation based on chitosan nanoparticles and green tea compounds has been investigated as a potential anti-cellulite agent [181].

Abosabaa et al. exploited the previously discussed properties of chitosan for the skin delivery of green tea extract [182]. The ion gelation method was employed to obtain CS–TPP stabilized by lecithin, which was used to increase the amount of extract retained by the colloidal structure [183]. Ex vivo permeation studies showed a significant accumulation of the extract components within the skin, confirming the effectiveness of the nanoformulation in improving the skin permeation of the tea derivatives. Additional studies performed on animal models demonstrated excellent anti-cellulite activity on subcutaneous adipocytes, thanks to the combined effects of both the nanosystems and green tea derivatives [184].

#### 6.4.2. Phenolic Acids

Phenolic acids are subclassified into benzoic acids and cinnamic acids [185]. A derivative of benzoic acid, gallic acid is a polyphenol known for its antioxidant, anti-inflammatory, and antimicrobial activities [186]. Due to these beneficial properties, it is used in cosmeceutical formulations for the treatment of skin disorders and as a cosmetic ingredient [186]. Shandil et al. developed chitosan NPs stabilized with Tween 80, containing gallic acid (GA) and rutin, to treat psoriasis [187]. In vitro tests showed that chitosan nanoformulations enhanced the effects of rutin and gallic acid, demonstrating that their combination significantly reduced the clinical signs of psoriasis, such as keratinocyte hyperproliferation and inflammation. Furthermore, the improved and faster penetration of the drugs through the skin was attributed to the surfactant effect of Tween 80, which facilitated the diffusion of the active compounds across the skin layers. This effect reduces the surface tension between molecules, enhancing the miscibility of the phenolic acids with the lipids of the stratum corneum. Moreover, it disorganizes the biobarrier, creating pathways that facilitate the penetration of the bioactives [187]. However, the low solubility of GA limits its direct application on the skin. In a recent study described by Abd-Elghany and Mohamad, ellagic acid (EA) was entrapped within chitosan-coated niosomes made up of Tween 80 and cholesterol, and their photoprotective effects on the HFB4 human skin fibroblasts and their ability to protect collagen from UV-induced degradation were evaluated. The obtained results revealed that the nanoformulations enhanced cell survival, increased the expression of Col1A1, TERT, and Timp3, and reduced the expression of MMP3, suggesting a protective effect on collagen and UV-exposed skin [188].

#### 6.4.3. Stilbenes

Resveratrol, a natural polyphenol with antioxidant, anti-inflammatory, anti-aging, cardioprotective, and neuroprotective properties, is a valuable compound found in red grapes and cranberries [189]. Sarma et al. encapsulated the molecule in anionic pectin-coated chitosan NPs to address issues related to resveratrol, such as its poor solubility, rapid metabolism, and low oral bioavailability. The developed core-shell nanoparticles efficiently retained the active compound, promoting a prolonged release for up to 30 h [190]. Chitosan nanoparticles containing resveratrol were also incorporated into hyaluronic acid-based hydrogels to create a topical formulation for the treatment of atopic dermatitis [189]. The nanoparticles were prepared via ion gelation using sodium tripolyphosphate (TPP) as an anionic crosslinking agent [191]. The CS:TPP ratio significantly influenced the nanoparticle size. Specifically, nanoparticles prepared with a chitosan:TPP ratio of 1:1 exhibited mean sizes of approximately 500 nm, while a ratio of 10:1 resulted in a notable decrease in size (~120 nm). All formulations displayed a positive surface charge, ranging from 13 to 20 mV, with the nanoparticles having a chitosan:TPP ratio of 10:1 showing the highest entrapment efficiency for resveratrol. The incorporation of the nanosystems into the hyaluronic acid matrix protected them from hydrolytic degradation and facilitated the slow release of the bioactive compound [189]. The resulting formulations effectively protected human keratinocytes (HaCaT) treated with TNF-α/INF-γ from ROS-induced damage and reduced the secretion of various pro-inflammatory molecules [191].

**Table 3 pharmaceutics-17-01068-t003:** Physico-chemical properties and applications of chitosan-based nanoparticles containing polyphenols.

Polyphenol	Composition and Characteristics	Biological Properties	Applications	Study Type	Results	Reference
Curcumin	Chitosan–TPP Mean diameter: ~30 to ~200 nm PdI: 0.129–0.536 ZP: /	Antioxidant and anti-inflammatory activity	Potential topical treatment for inflammatory skin conditions	Ex vivo/ in vitro	Enhanced skin penetration by transdermal pathways demonstrated via confocal microscopy; targeted follicular localization acting as drug reservoirs sustained drug release	[165]
Curcumin	Chitosan–TPP Mean diameter: 167.3 nm (1:3) to 251.5 nm (1:5) PdI: ~0.45 ZP: 18–20 mV	Antioxidant, anti-inflammatory, and antimicrobial activity	Cancer, psoriasis, hypertrophic scars, vitiligo, wound healing	In vitro	Uniform size, enhanced permeation, controlled release, good cell compatibility	[169]
Quercetin	Chitosan–lecithin Mean diameter: ~95 nm PdI: ~0.45 ZP: ~11 mV	Antioxidant and anti- inflammatory activity	Acne, rosacea	Ex vivo/ in vitro	Enhanced skin permeation and higher epidermal accumulation compared to the free quercetin solution.	[171]
Quercetin	Chitosan Mean diameter: ~184 nm PdI: / ZP: 37 mV	Antioxidant, UVB radiation damage protection	Skin aging, skin damage	In vitro/ in vivo	Reduced inflammation and oxidative stress, UVB protection	[174]
Epigallocatechin gallate (EGCG)	Chitosan Mean diameter: ~200 nm PdI: 0.13 ZP: ~38 mV	Antioxidant, anti-inflammatory, and antimicrobial activity	Psoriasis	In vitro/ in vivo	Anti-inflammatory effects, improved psoriatic symptoms	[180]
Green Tea Extract (GTE)	Chitosan Mean diameter: ~300 nm PdI: ~0.25 ZP: ~41 mV	Antioxidant, anti-inflammatory, and antimicrobial activity	Cellulitis	Ex vivo/ in vitro	Reduced adipocyte size and skin thickness; effective anti-cellulite action	[181]
Gallic acid/Rutin	Chitosan Mean diameter: 300–350 nm PdI: <1.0 ZP: /	Antioxidant, anti-inflammatory, and antimicrobial activity	Psoriasis	In vitro	Inhibition of hyperproliferation, oxidative stress, and inflammation	[187]
Ellagic acid	Chitosan-coated Niosomes Mean diameter: ~53 nm PdI: ~0.87 ZP: /	UVB radiation damage protection	Skin wrinkles and photoaging	In vitro	Anti-aging activity, protection against UV damage	[188]
Resveratrol	Chitosan Mean diameter: ~120 nm PdI: ~0.24 ZP: ~19 mV	Antioxidant and anti-inflammatory activity	Atopic dermatitis	In vitro	Decreased oxidative stress and inflammation, potential coadjuvant treatment	[189]

PdI: Polydispersity index; ZP: Zeta potential.

## 7. Conclusions

Consumers are increasingly moving away from synthetic chemicals in cosmetic products in favor of natural alternatives, which have fewer side effects and are more environmentally friendly. Therefore, natural plant extracts or plant-derived by-products can be used to develop natural formulations with pharmacological activity, maximizing the potential of underutilized or discarded compounds. In the last decade, several experimental investigations have focused on plant extract-based cosmeceuticals, and the number of such studies is expected to rise given the abundance of natural molecules with a wide range of attractive properties for beauty and health [192]. Polyphenols have been shown to exert various beneficial biological effects on the skin, making them ideal candidates for use in human skin care [193]. Despite their multifunctional properties, they have some limitations related to their physico-chemical properties and biomaterials such as chitosan have been employed to develop novel cosmeceuticals capable of addressing these issues.

The data discussed herein show that the treatment of skin diseases with polyphenol-loaded chitosan nanoparticles may be a novel and promising therapeutic approach. To the best of the authors’ knowledge, there are no formulations based on the use of this type of nanoparticles currently available on the market. The experimental investigations described in this review provide the rationale for their use in the proposed applications and demonstrate their potential impact on skin healthcare. However, it should be noted that despite the growing interest in this technology, several critical issues must be addressed. Firstly, the majority of experimental investigations are focused on in vitro evaluations or conducted on animal models, while there is still a lack of solid and well-designed clinical data to confirm efficacy and safety in humans [155]. Moreover, the development of standardized formulation protocols is essential to ensure reproducibility and scalability. Key aspects such as the long-term stability of nanoparticles, their ability to effectively penetrate the skin barrier, and the controlled release of polyphenols under physiological conditions remain insufficiently explored. There is also limited attention to long-term effects and potential interactions with other skin components, as well as to conceivable risks of immunogenicity or skin irritation [149]. Finally, only a few polyphenols have been entrapped within nanoparticles, and their chemical complexity needs additional studies to refine the formulations and ensure the maximum bioactivity of the compounds. As previously reported, cosmeceuticals based on chitosan nanoparticles are not yet available on the market. The challenges include regulatory approval processes, large-scale production at sustainable costs, and maintaining product consistency. Addressing these issues represents a crucial step toward the future commercialization of the described nanocosmeceuticals.

## Figures and Tables

**Figure 1 pharmaceutics-17-01068-f001:**
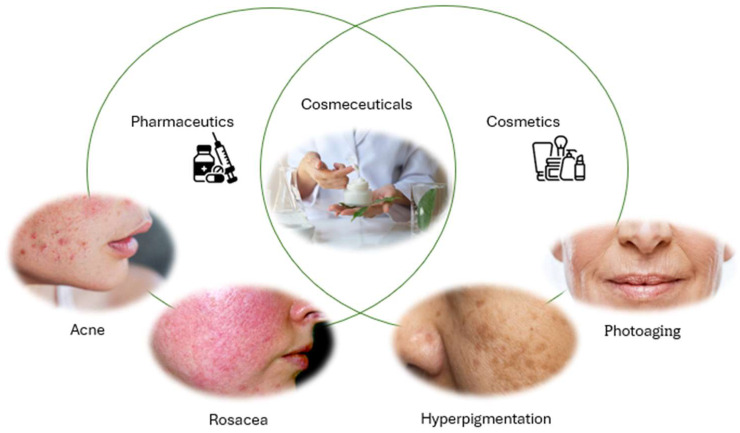
Graphical representation of some cosmeceutical applications.

**Figure 2 pharmaceutics-17-01068-f002:**
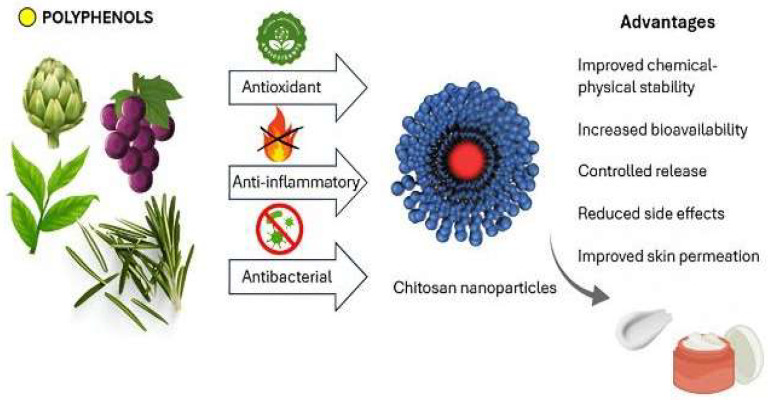
Advantages of the use of chitosan nanoparticles as drug carriers of polyphenols through the skin.

**Figure 3 pharmaceutics-17-01068-f003:**
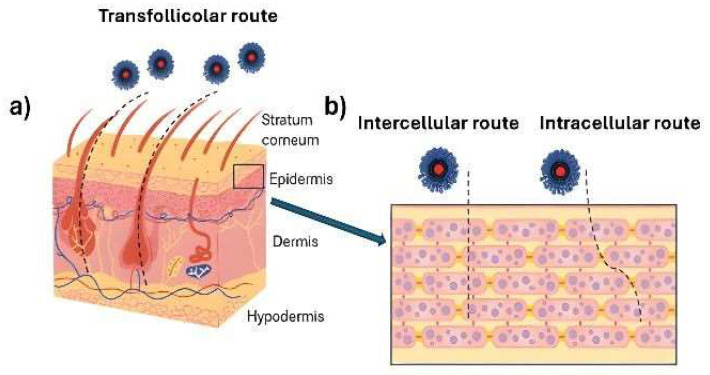
Schematic representation of the passage of nanoparticles through (**a**) the sweat glands/hair follicles and (**b**) through the stratum corneum.

**Figure 4 pharmaceutics-17-01068-f004:**
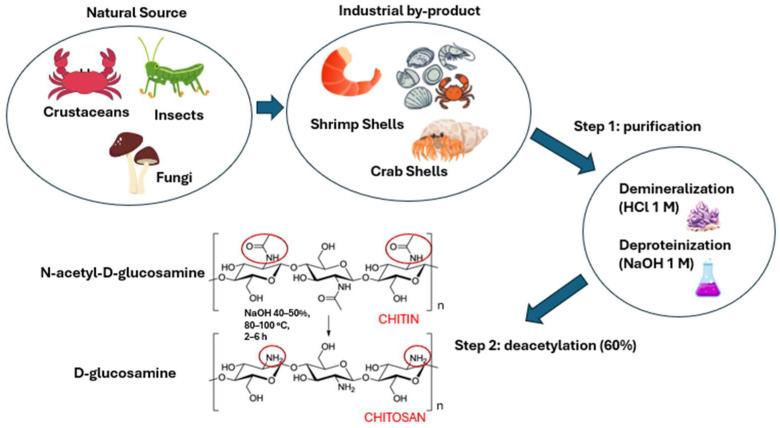
Schematic representation of natural and industrial sources of chitin and its conversion into chitosan.

**Figure 5 pharmaceutics-17-01068-f005:**
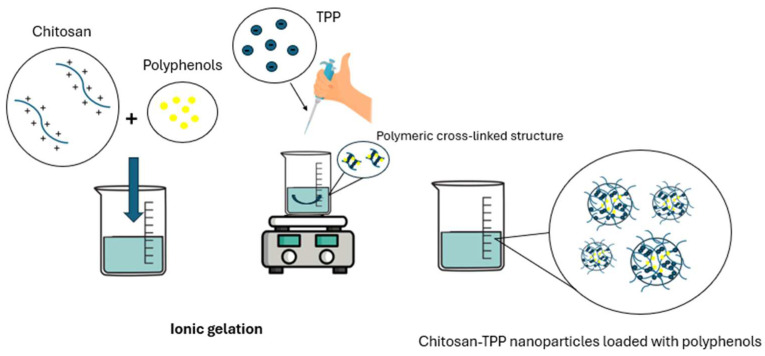
Schematic representation of the development of chitosan–TPP nanoparticles containing polyphenols obtained by ionic gelation.

**Table 1 pharmaceutics-17-01068-t001:** Classification of polyphenols and their subclasses.

Polyphenol Class	Subclasses	Description	Common Examples	Main Properties
Flavonoids	Flavonols	Flavonoids with a 3-hydroxyflavone structure	Quercetin, curcumin	Antioxidant, anti-inflammatory, UV protection, antimicrobial activity
	Flavones	Flavonoids lacking the 3-hydroxy group	Apigenin, luteolin	Antioxidant, anti-inflammatory, antimicrobial activity
	Flavanones	Saturated flavonoids	Naringenin, hesperetin	Antioxidant, anti-inflammatory
	Flavanols (Catechins)	Flavonoids with hydroxyl groups	Epigallocatechin gallate (EGCG)	Antioxidant, antimicrobial
	Anthocyanidins	Pigmented flavonoids	Cyanidin	Antioxidant, pigment properties
	Isoflavones	Flavonoids with B-ring attached at position 3	Genistein	Phytoestrogenic, antioxidant
Phenolic Acids	Hydroxybenzoic acids	Phenolic acids with a C6–C1 structure	Gallic acid,	Antioxidant, anti-inflammatory
	Hydroxycinnamic acids	Phenolic acids with a C6–C3 structure	Ferulic acid, caffeic acid	Antioxidant, anti-inflammatory
Stilbenes		Compounds with two aromatic rings linked by ethylene bridge	Resveratrol	Antioxidant, anti-inflammatory, cardioprotective
Lignans		Dimers of phenylpropanoids	Secoisolariciresinol	Antioxidant, estrogen-like effects

**Table 2 pharmaceutics-17-01068-t002:** Comparison of methods employed to develop chitosan nanoparticles.

Method	Principle	Advantages	Disadvantage	Cosmetic/Pharmaceutical Applications	References
Ionic gelation	Crosslinking with polyanionic agents (e.g., TPP)	Simple, fast, no organic solvents, room temperature	Limited stability if not optimized, particle size > 100 nm	Encapsulation of polyphenols and essential oils (anti-aging, antioxidant)	[156,157]
Polyelectrolyte complexation	Electrostatic complex formation between oppositely charged polymers without organic solvents	Free of organic solvents	Less reliable particle size and stability control depending on materials	Usable but with limitations on quality control	[161]
Emulsification and solvent evaporation	Formation of emulsions with organic solvents, followed by solvent evaporation to obtain nanoparticles	Production of nanoparticles	Use of organic solvents and surfactants that may be irritating or toxic; less precise size control	Less suitable for delicate applications such as cosmetics	[156]
Microemulsion	Formation of thermodynamically stable emulsions containing organic solvents to produce very small nanoparticles	Very small nanoparticles (<100 nm)	Use of organic solvents; longer processing times	Produces very small nanoparticles but less practical for cosmetics	[156,162]
Chemical crosslinking	Chemical crosslinking of chitosan using agents like glutaraldehyde	Stable nanoparticles	Use of agents such as glutaraldehyde that may leave undesirable residues; compromises biological properties	Unsuitable for cosmetic use	[163]
Complex coacervation	Phase separation of a polymer-rich phase in aqueous environment at low temperature	Aqueous environment at low temperature	Sensitive to pH variations, causing destabilization or ineffective encapsulation	Limited by the need to maintain pH compatible with skin	[164]

## Data Availability

No primary research results, software, or code have been included, and no new data were generated or analysed as part of this review.

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
