# Peer review of "Chitosan Nanoparticles Loaded with Polyphenols for Cosmeceutical Applications: A State-of-the-Art Review"

_pharmaceutics, 2025, doi:10.3390/pharmaceutics17081068_

Round 1

Reviewer 1 Report (Previous Reviewer 3)

Comments and Suggestions for Authors

All my concerns have been addressed. Thus, an acceptance is suggested.

Author Response

Reviewer 1: All my concerns have been addressed. Thus, an acceptance is suggested.

Authors : We thank the Reviewer for the valuable feedback and positive evaluation.

Reviewer 2 Report (Previous Reviewer 5)

Comments and Suggestions for Authors

Redundant phrases between lines 22 and 27: the idea that systems increase stability and bioavailability is repeated.

Line 24 contains a punctuation error ("properties. .").

Expressions such as "This review will discuss..." could be rephrased in the active voice (e.g., "This review discusses...").

Introduction:There is repetition of concepts such as the effects of aging, oxidative stress, and the impact of free radicals.

Paragraphs 85–101 could be condensed to improve readability and avoid redundancy.

The authors cite "conventional formulations" and "several studies" without proper references. These should be specified or clarified with citations.

"This review aims to..." appears more than once (lines 162 and 165) in almost identical form—remove or rephrase one to avoid repetition.

Figure 3 could be better referenced in the text with an additional explanation of the role of hair follicles as reservoirs in topical delivery.

Lines 292–439):There are content repetition issues between the studies on resveratrol, green tea, etc. Redundant results and mechanisms should be streamlined.

Some information is repeated using different phrasing but the same references—this may lead to self-plagiarism. Consider consolidating where appropriate.

Consolidate the studies on resveratrol (lines 303–324 and 315–323) into a single, more critical paragraph, highlighting formulation variations and distinct cellular mechanisms.

The explanation on ionic gelation is repeated in different parts of the manuscript. It should appear once, with any subsequent mention referring back to the initial explanation.

“Chitosan being a cationic polymer that exhibits protonated amino groups under physiological conditions…” (line 553): This sentence should be reworded to avoid ambiguity and grammatical errors. Suggested: "As a cationic polymer, chitosan exhibits protonated amino groups under physiological conditions, which..."

6.1 Flavonoids: There are several repetitions between the studies on quercetin and curcumin. It is possible to summarize and compare their outcomes directly (e.g., release profile, entrapment efficiency, skin permeation).

Author Response

The Authors are very grateful to the Reviewer for the valued queries and advice. It is the opinion of the authors that the following changes in the manuscript have improved the quality of the paper. A response to each point raised in the main text has been shown in red.

Reviewer 2 (R2): Redundant phrases between lines 22 and 27: the idea that systems increase stability and bioavailability is repeated.

Authors (A): According to the Reviewer’s suggestion, the phrases between lines 22 and 27 have been duly revised.

R2: Line 24 contains a punctuation error ("properties. .").

A: We thank the Reviewer; the typo on line 24 (“properties. .”) has been corrected.

R2: Expressions such as "This review will discuss..." could be rephrased in the active voice (e.g., "This review discusses...").

A: As requested by the Reviewer, the expression “this review will discuss..”" has been rephrased in the active voice “This review discusses..” (line 20).

R2: Introduction: There is repetition of concepts such as the effects of aging, oxidative stress, and the impact of free radicals.

A: In response to the Reviewer’s comment, the text has been duly revised.

R2: Paragraphs 85–101 could be condensed to improve readability and avoid redundancy.

A: According to the Reviewer’s suggestion, the section has been duly revised and condensed to improve clarity and eliminate redundancies.

R2: The authors cite "conventional formulations" and "several studies" without proper references. These should be specified or clarified with citations.

A: In response to the Reviewer’s comment, the citations have been added (lines 91 and 106).

R2: "This review aims to..." appears more than once (lines 162 and 165) in almost identical form—remove or rephrase one to avoid repetition.

A: We thank the Reviewer for this suggestion. The text has been duly revised (new lines 160 and 164).

R2: Figure 3 could be better referenced in the text with an additional explanation of the role of hair follicles as reservoirs in topical delivery.

A: We thank the Reviewer for this suggestion. In response, we have revised the manuscript to better integrate Figure 3 into the text.

R2: Lines 292–439): There are content repetition issues between the studies on resveratrol, green tea, etc. Redundant results and mechanisms should be streamlined.

A: In response to the Reviewer’s comment, we carefully reviewed the text and reorganized the content to eliminate redundancies and improve the flow of the section (lines 302-466).

R2: Some information is repeated using different phrasing but the same references—this may lead to self-plagiarism. Consider consolidating where appropriate.

A: In response to the Reviewer’s comment, we have revised the manuscript by removing redundant content to improve clarity and avoid potential self-plagiarism.

R2: Consolidate the studies on resveratrol (lines 303–324 and 315–323) into a single, more critical paragraph, highlighting formulation variations and distinct cellular mechanisms.

A: We thank the Reviewer for this suggestion. We revised the paragraph to make it clearer and more critical, highlighting the differences of the various formulations and the cell mechanisms involved. However, we preferred to keep the content within the paragraph where it was originally discussed in order to ensure a suitable flow and coherence (lines 313–333).

R2: The explanation on ionic gelation is repeated in different parts of the manuscript. It should appear once, with any subsequent mention referring back to the initial explanation.

A: In response to the Reviewer’s comment, the explanation of the ionic gelation process is now provided only once in the manuscript (line 636), with subsequent mentions referring back to this initial description (line 644).

R2: “Chitosan being a cationic polymer that exhibits protonated amino groups under physiological conditions…” (line 553): This sentence should be reworded to avoid ambiguity and grammatical errors. Suggested: "As a cationic polymer, chitosan exhibits protonated amino groups under physiological conditions, which..."

A: We thank the Reviewer for this suggestion. The sentence has been revised as recommended and it is now included at line 581.

R2: 6.1 Flavonoids: There are several repetitions between the studies on quercetin and curcumin. It is possible to summarize and compare their outcomes directly (e.g., release profile, entrapment efficiency, skin permeation).

A: According to the Reviewer’s suggestion, section 6.1 Flavonoids has been duly revised and is now presented as section 6.3.1.

Reviewer 3 Report (New Reviewer)

Comments and Suggestions for Authors

This work summarizes the application of polyphenol drugs as chitosan-loaded substances in the field of cosmeceuticals in recent years. The author explains the development and advantages of chitosan as a drug delivery platform by discussing the health problems of the skin and the disadvantages of polyphenol drugs in addressing these problems. This manuscript has certain research value and can be considered for publication after some revisions:

1) There are many red-marked corrections in the manuscript, which affect the review.

2) Under the fifth part “Chitosan in cosmetics and cosmeceuticals” and sixth part “Chitosan nanoparticles as carriers of polyphenols for topical applications”, the description of chitosan needs to be reorganized and simplified, the current organization is not good.

3) There is a minor issue with the citation format of the literature, such as [18, 19, 20, 21] should be written as [18-21]. Please review the entire manuscript.

Author Response

The Authors are very grateful to the Reviewer for the valued queries and advice. It is the opinion of the authors that the following changes in the manuscript have improved the quality of the paper. A response to each point raised in the main text has been shown in red.

Reviewer 3 (R3): There are many red-marked corrections in the manuscript, which affect the review.

Authors (A): The red-marked corrections in the manuscript were included in response to a previous of revision as required by the EO. In the revised manuscript only the new changes are highlighted.

R3: Under the fifth part “Chitosan in cosmetics and cosmeceuticals” and sixth part “Chitosan nanoparticles as carriers of polyphenols for topical applications”, the description of chitosan needs to be reorganized and simplified, the current organization is not good.

A: As requested by the Reviewer, the description of chitosan in the fifth and sixth sections of the text has been reorganized and simplified.

R3: There is a minor issue with the citation format of the literature, such as [18, 19, 20, 21] should be written as [18-21]. Please review the entire manuscript.

A: In response to the Reviewer’s comment, we have corrected all the citations in the manuscript.

Round 2

Reviewer 2 Report (Previous Reviewer 5)

Comments and Suggestions for Authors

Figure 4 lacks clarity and lacks in-depth discussion. The natural sources of chitin (crab, mushroom, and insect) are loosely arranged, with no explicit connections to the chemical structures shown below. Furthermore, the figure does not adequately represent the main industrial sources of chitin, shrimp shells, for example, are widely used industrially but are not included.

lines 502–508: It would be important to include a brief but more detailed description of the deacetylation process. The text only mentions the removal of proteins and acetyl groups, but does not specify how this step is performed (the use of concentrated NaOH, the temperature range, and the reaction time), which has a direct impact on the final properties of chitosan, such as the degree of deacetylation and solubility, MM,...

At acidic pH, the amino groups can be protonated..." is correct, but it would be even clearer to specify "pKa ≈ 6.3" to contextualize chitosan's behavior in solutions with different pH values.

The chelating property is mentioned only at the end of the paragraph and very briefly. Considering its relevance in environmental, biomedical, and pharmaceutical applications, it is recommended to expand on this point a bit or connect it with possible applications mentioned previously.

The term "solubility" is repeated several times. Variations such as "aqueous dispersion capacity" or "dissolution behavior" are suggested to make the text more fluid.

"Marine fungi" would be more accurate, as chitin is present in the cell walls of fungi in general, not just marine fungi.

"Removing proteins and at least 60% of its acetyl groups" - how? Protein removal occurs in the chitin purification step, before deacetylation. what happens in deacetylation???

Author Response

The Authors are very grateful to the Reviewer for the valued queries and advice. It is the opinion of the authors that the following changes in the manuscript have improved the quality of the paper. A response to each point raised in the main text has been shown in red.

Reviewer 2 (R2): Figure 4 lacks clarity and lacks in-depth discussion. The natural sources of chitin (crab, mushroom, and insect) are loosely arranged, with no explicit connections to the chemical structures shown below. Furthermore, the figure does not adequately represent the main industrial sources of chitin, shrimp shells, for example, are widely used industrially but are not included.

Authors (A): We thank the reviewer for the suggestion. We have revised Figure 4 by including the requested chitin sources, such as shrimp shells, and reorganized it to improve clarity and explicitly connecting the natural sources to the chemical structures. Additionally, we have clarified the figure caption and expanded the discussion in the text to provide a more thorough explanation (section 6).

R2: lines 502–508: It would be important to include a brief but more detailed description of the deacetylation process. The text only mentions the removal of proteins and acetyl groups, but does not specify how this step is performed (the use of concentrated NaOH, the temperature range, and the reaction time), which has a direct impact on the final properties of chitosan, such as the degree of deacetylation and solubility, MM,...

A: We thank the reviewer for the comment. The revised manuscript includes a more detailed description of the deacetylation process. The revised paragraph can be found at lines 488-510.

R2: At acidic pH, the amino groups can be protonated..." is correct, but it would be even clearer to specify "pKa ≈ 6.3" to contextualize chitosan's behavior in solutions with different pH values.

A: We thank the reviewer for the comment. We have duly revised the text (line 497).

R2: The chelating property is mentioned only at the end of the paragraph and very briefly. Considering its relevance in environmental, biomedical, and pharmaceutical applications, it is recommended to expand on this point a bit or connect it with possible applications mentioned previously.

A: We thank the reviewer for the comment. We have duly revised the text (lines 511-516).

R2: The term "solubility" is repeated several times. Variations such as "aqueous dispersion capacity" or "dissolution behavior" are suggested to make the text more fluid.

A: We thank the reviewer for the suggestion. We have revised the text accordingly to improve clarity and variation in terminology.

R2: "Marine fungi" would be more accurate, as chitin is present in the cell walls of fungi in general, not just marine fungi.

A: We thank the reviewer for the comment. We have duly revised the text (line 483).

R2: "Removing proteins and at least 60% of its acetyl groups" - how? Protein removal occurs in the chitin purification step, before deacetylation. what happens in deacetylation???

A: We thank the reviewer for the comment. We have duly revised the text (lines 485–494).

A: The language has been reviewed by a native speaker of American English.

Reviewer 3 Report (New Reviewer)

Comments and Suggestions for Authors

This work summarizes the application of polyphenol drugs as chitosan-loaded substances in the field of cosmeceuticals in recent years. The author explains the development and advantages of chitosan as a drug delivery platform by discussing the health problems of the skin and the disadvantages of polyphenol drugs in addressing these problems. This manuscript has certain research value and can be considered for publication after minor revisions. The following issues can be optimized and modified:

1) There are many red-marked corrections in the manuscript, which affect the review.

2) Under the fifth title “Chitosan in cosmetics and cosmeceuticals” and the sixth title “Chitosan nanoparticles as carriers of polyphenols for topical applications”, there are redundant and confusing descriptions of chitosan.

3)The arguments mentioned in the article should be consistent with the content of the cited literature. For instance, in Table 3, Physico-chemical properties and applications of chitosan-based nanoparticles containing polyphenols, the biological properties, applications, and results of curcumin do not align with the content of the cited literature, “Tracking the Transdermal Penetration Pathways of Optimized Curcumin-Loaded Chitosan Nanoparticles via Confocal Laser Scanning Microscopy”.

4) Under the title “Chitosan Nanoparticles as Carriers of Polyphenols for Topical Applications”, the methods for preparing chitosan nanoparticles should be briefly summarized, with emphasis placed on highlighting the application of polyphenols combined with chitosan.

5) Suggest adding images of typical examples and corresponding mechanisms to enhance the readability of the review.

Author Response

The Authors are very grateful to the Reviewer for the valued queries and advice. It is the opinion of the authors that the following changes in the manuscript have improved the quality of the paper. A response to each point raised in the main text has been shown in red.

Reviewer 3 (R3): There are many red-marked corrections in the manuscript, which affect the review.

Authors (A): The red-marked corrections in the manuscript were included in response to a previous request from the journal, aimed at facilitating the evaluation of the revisions made following the earlier round of reviewers’ comments. Additionally, a clean version of the manuscript has been provided. However, the parts currently highlighted in the text are due to the new round of revision.

R3: Under the fifth title “Chitosan in cosmetics and cosmeceuticals” and the sixth title “Chitosan nanoparticles as carriers of polyphenols for topical applications”, there are redundant and confusing descriptions of chitosan.

A: We thank the reviewer for the observation. We have carefully revised and reorganized Sections 5 and 6 in order to reduce redundancies and improve clarity.

R3: The arguments mentioned in the article should be consistent with the content of the cited literature. For instance, in Table 3, Physico-chemical properties and applications of chitosan-based nanoparticles containing polyphenols, the biological properties, applications, and results of curcumin do not align with the content of the cited literature, “Tracking the Transdermal Penetration Pathways of Optimized Curcumin-Loaded Chitosan Nanoparticles via Confocal Laser Scanning Microscopy”.

A: We thank the reviewer for this observation. We have carefully reviewed the cited article “Tracking the Transdermal Penetration Pathways of Optimized Curcumin-Loaded Chitosan Nanoparticles via Confocal Laser Scanning Microscopy” and have accordingly updated Table 3 to ensure full consistency and accuracy with the text.

R3: Under the title “Chitosan Nanoparticles as Carriers of Polyphenols for Topical Applications”, the methods for preparing chitosan nanoparticles should be briefly summarized, with emphasis placed on highlighting the application of polyphenols combined with chitosan.

A: We thank the reviewer for this comment. We have revised the section titled “Chitosan Nanoparticles as Carriers of Polyphenols for Topical Applications,” focusing specifically on the methods for preparing chitosan nanoparticles to improve clarity and communicative effectiveness. Additionally, we have placed greater emphasis on the specific advantages of combining polyphenols with chitosan using the ionic gelation method.

R: Suggest adding images of typical examples and corresponding mechanisms to enhance the readability of the review.

A: We thank the reviewer for this suggestion. We have added an image (Figure 5) illustrating the mechanism preparation of chitosan nanoparticles containing polyphenols in order to enhance the readability and clarity of the text.

 A: The language has been reviewed by a native speaker of American English.

This manuscript is a resubmission of an earlier submission. The following is a list of the peer review reports and author responses from that submission.

Round 1

Reviewer 1 Report

Comments and Suggestions for Authors

The review article presents an overview on the role of chitosan nanoparticles loading polyphenolic compounds as cosmeceuticals, which is of particular interest to readers in the field of cosmetics, formulation and natural compounds. The review article is interesting to read, and addresses a knowledge gap of integration of nanotechnology in cosmeceutical industries. References are overall adequate. However, some comments need to be addressed:

1- Figure 1, please remove psoriasis from the figure as it is a dermatological disease which does not really qualify as target for cosmeceutical applications. The authors can alternatively mention rosacea, as well as dry/compromised skin barrier in general

2- Section 5, the mention of polymeric nanoparticles especially nanocapsules part is irrelevant, since chitosan nanoparticles are matrix-based nanoparticles, and the review article specifically focuses on them even in the title. I suggest removing this part.

3- Table 1, add a column which defines the status of investigation of each paper (in vitro, in vivo, or clinical)

4- Before the conclusion section, the authors need to highlight the limitations/challenges/commercialization status and clinical status of chitosan nanoparticles for cosmeceutical purposes.

Reviewer 2 Report

Comments and Suggestions for Authors

Although a review article like this may result in some additional citations for the journal, it should not be accepted for publication unless it has some new and significant information to add to this frenetic reviewing activity. A good review includes a critical evaluation of the field based on the most significant original publication. However, this requires a significant expertise, and thus, the reviewer must have a good understanding of the field. The author is of this review is, apparently, a co-author of just a few recent publications, where Chitosan nanoparticles is a topic. One of the key features of this review is that it is mainly based on other review articles, many of which are also written by non-experts. Such reviews, written by non-experts and partially based on reviews by other non-experts, is unlikely to correct any errors or misconceptions and is actually likely to enhance them and possibly add some more. This review is well organized and the outline is correct. However, the starting hypothesis, novelty, and significance of review are unclear. Also, they have not been successful in presenting the work to the extent that excites the reader towards such a study. Therefore, major issues have to be addressed by authors to reconsider. Some of the comments regarding the submitted manuscript are provided below:

General comments

  1. The authors are invited to include a paragraph approaching the main contributions of their study. If there are other similar reviews, you must mention and highlight the specific updates provided by your research.
  2. The abstract cannot fully explain the content of this manuscript. Make the abstract very concise and make the novelty part very strong.
  3. In introduction part; it is very poor in up to date citation regarding the use of Chitosan nanoparticles for encapsulation bioactive materials. Please address this comment in light of the following articles in the introduction part: https://doi.org/10.1016/j.ijbiomac.2025.145123; https://doi.org/10.1007/s00289-025-05650-5; https://doi.org/10.1016/j.ijbiomac.2023.124985; https://doi.org/10.1016/j.jddst.2023.104388; https://doi.org/10.3390/pharmaceutics15010081; https://doi.org/10.3390/pharmaceutics14071350; https://doi.org/10.1016/j.ijbiomac.2020.10.153) to give the reader more insights regarding the the use of natrual material for encapsulation bioactive compound and their biocompatibility for many applications as multifunctional smart materials.
  4. It is necessary for the author to improve the writing of the manuscript. This will help to improve the readability of this manuscript.
  5. In general, there are many grammatical and typographical errors and few sentences are inconsistent and incomprehensible; the authors have to check the whole manuscript carefully to eliminate all editing errors.

The whole manuscript and a complete proofreading is needed. The whole paper review should be revised by a native English speaking scientist and professional editing program.

  1. This review must be completely rewritten with a clear focus and some discussion to show that it is actually providing some new and valuable information regarding the protocols used for the preparation (with Schemes) and physicochemical properties of the obtained nano- and micro-formulation.
  2. How does your paper contribute to the advancement of knowledge?
  3. Morphology-application correlations should be established.
  4. The conclusion section should be expanded to express the study's conclusions more clearly and include suggestions for future research.
  5. No link is offered to the future directions in this area of research; therefore, the manuscript looks somewhat unfinished. What are the future courses of this work? How this research can be advanced further? What is your vision of future directions? A short discussion of these points should be added to the "CONCLUSON" section

Specific comments:

  1. It is necessary for the authors to standardize the image format, and the quality of multiple figures is low, the authors need to improve the resolution of the figures as much as possible.
  2. Please make the conclusion brief and discuss the main understandings from the review conducted.
  3. Incorporating a more forward-looking perspective on emerging trends and future directions would enhance the relevance of the review.
  4. The manuscript only needs to explain the effect of Chitosan nanoparticles on skin and whether the structural modification of chitosan is appropriate

Comments on the Quality of English Language

The English should be improved to more clearly express the research.

Reviewer 3 Report

Comments and Suggestions for Authors

In this manuscript, the authors provide an overview of the recent progresses on the polyphenols-incorporated chitosan nanoparticles for the cosmeceutical applications. In general, this review demonstrates well-discussed topics with a high quality, which may appeal to a broad readership in the field of skincare using chitosan or polyphenolic related biomaterials. While some questions remain to be addressed.

  1. What does the phrase “topic administration” mean shown as one of the Keywords in the manuscript?
  2. Figure 3b shows two different pathways of nanoparticles passing through the stratum corneum. But in the main text, no explanations were given. More discussions about their differences and principles of material design should be provided.
  3. What is the driving force of polyphenolic compounds or natural extracts incorporating to chitosan nanoparticles?
  4. Why does the nanoparticle formulation of bioactive ingredients permeate more effectively than their free form?
  5. What is the dosage form of polyphenols-loaded chitosan nanoparticles used in cosmeceuticals?

Reviewer 4 Report

Comments and Suggestions for Authors

Please consider the following comments during the revision process.

  1. Please justify the rationale behind the manuscript.
  2. The introduction is too general and lacks focus on the selected topic. The manuscript provides mostly general information, with only minor emphasis on aspects of chitosan nanoparticle formulation. Please justify this.
  3. The manuscript should highlight the key points related to the title without losing sight of the overall focus.

Reviewer 5 Report

Comments and Suggestions for Authors

The manuscript entitled “Chitosan nanoparticles containing polyphenols for cosmeceutical application: a state of the art” presents a relevant and current review. It is well detailed and well connected to the subject. Therefore, I believe that in order to be accepted, some improvements should be made, such as:

Suggest making it more appealing and better aligned with the field’s terminology.
Suggested title: “Chitosan Nanoparticles Loaded with Polyphenols for Cosmeceutical Applications: A State-of-the-Art Review”.

Recommend making the text more direct and cohesive. For example, generic expressions such as “generated significant interest” could be replaced with more specific and impactful phrases.

It is recommended to reorganize some excerpts, such as the paragraph on line 83 (“As a neuroendocrine-immune organ...”), to better connect with the objectives of the review.

The explanation of the structure and classification of polyphenols is good, but I believe that using a table would help summarize the main classes and improve visualization. The subsections (3.1–3.3) are comprehensive but could be more concise. Some examples are repeated and could be organized in a summary table. For example, compounds such as resveratrol, green tea catechins (especially EGCG), gallic acid, and citrus extracts are mentioned in different sections with similar information. Resveratrol is described separately as an antioxidant and anti-inflammatory, but both effects are part of the same multifunctional mechanism and could be discussed in a unified way. Similarly, green tea is discussed in 3.1 for its effects against oxidative stress and in 3.3 for its anti-acne action, based on the same clinical study, which results in redundancy. Citrus extracts are mentioned for both their antioxidant and anti-inflammatory properties, with a focus on photoprotection and COX-2 modulation. Gallic acid is also mentioned more than once with different emphasis.

I suggest including at least one comparative table summarizing: type of polyphenol × type of nanoparticle × therapeutic application × physicochemical characteristics × results. Table 1 should include units, a review of PdI and ZP values, and uniform formatting.

It would also be important to emphasize the advantages of the ionic gelation method, especially in lines 114–115 and 465–466, where the technique is only briefly mentioned. Although it is described as simple, non-toxic, and free of organic solvents, direct comparisons with other approaches (such as emulsification, chemical crosslinking, or polyelectrolyte complexation) are missing. These comparisons could help highlight why ionic gelation is more suitable for cosmetic applications, due to factors such as greater biocompatibility, lower risk of toxic residues, and better compatibility with sensitive active compounds like polyphenols. Furthermore, between lines 466–469, where the impact of nanoparticle size on skin permeation is discussed, a direct link could be made with the advantages of ionic gelation in producing controlled-size particles with low polydispersity and good colloidal stability.

Very long sentences should be rewritten throughout the manuscript. There are passages with more than 30 words, such as in lines 230–240 and 374–380. Additionally, the repetitive use of expressions such as “It has been demonstrated that…” should be reduced.

The conclusion could be improved by adding a more critical view of research gaps and future perspectives (e.g., “need for clinical validation”, “standardized formulation protocols”). The expression “To the best of our knowledge…” could also be replaced with a more cautious and scientifically neutral formulation.

Reviewer 6 Report

Comments and Suggestions for Authors

I consider the manuscript number pharmaceutics-3733126, entitled "Chitosan nanoparticles containing polyphenols for cosmeceutical application: a state of the art" is suitable for publication in this journal.

This manuscript is well structured, starting with the basics of the formulations, the active principles and ending with the state of the art of chitosan nanospheres for this application. The references are well chosen and commented.